Characteristic of persistent human papillomavirus infection in women worldwide: a meta–analysis

Zhao Ming 1 2
Zhou Dan 1 2
Zhang Min 1 2
Kang Peipei 3
Cui Meimei 2 4
Zhu Liling jmsuyf@163.com 1
Luo Limei jmsllm@163.com 2
1 School of Public Health, Jiamusi University , Jiamusi , Heilongjiang , China
2 Maternal and Child Health Development Research Center, Shandong Provincial Maternal and Child Health Care Hospital Affiliated to Qingdao University , Jinan , Shandong , China
3 Shandong Mental Health Center, Shandong University , Jinan , Shandong , China
4 School of Basic Medical, Weifang Medical College , Weifang , Shandong , China
Jose Leny
Electronic publication date: 2023 Nov 14
Publication date: 2023
Volume: 11
Electronic Location ID: e16247
Received 2023 Jul 28; Accepted 2023 Sep 14
Copyright: ©2023 Zhao et al.
Copyright year: 2023
Copyright holder: Zhao et al.
License: This is an open access article distributed under the terms of the Creative Commons Attribution License, which permits unrestricted use, distribution, reproduction and adaptation in any medium and for any purpose provided that it is properly attributed. For attribution, the original author(s), title, publication source (PeerJ) and either DOI or URL of the article must be cited.
License URL: https://creativecommons.org/licenses/by/4.0/

Keywords: Human papillomavirus, Persistent infection, Genotype, Meta-analysis, HPV

Funding: The Shandong Humanities and Social Science Project 2021-ZXJK-14 Shandong Provincial Maternal and Child Health Care Hospital Project 2021SFF058 This study received funding from the Shandong Humanities and Social Science Project (2021-ZXJK-14) and the Shandong Provincial Maternal and Child Health Care Hospital Project (2021SFF058). The funders had no role in study design, data collection and analysis, decision to publish, or preparation of the manuscript.

==============================
Objectives

We aimed to estimate the genotype distribution of persistent human papillomavirus (HPV) infection in females worldwide, and provided a scientific basis for the prevention strategies of cervical cancer (CC) and the development of HPV vaccines.

Methods

Both English and Chinese databases were researched from the inception to July 2023. The pooled persistent HPV infection prevalence was calculated using a random effects model. The subgroup analysis was performed to explore the heterogeneity. Publication bias was evaluated using funnel plot, Egger’s and Begg’s test.

Results

Twenty-eight studies with 27,335 participants were included. The pooled prevalence of persistent HPV infection was 29.37% (95% CI [24.05%∼35.31%]), and the genotypes with the persistent infection prevalence were HPV16 (35.01%), HPV52 (28.19%), HPV58 (27.06%), HPV18 (25.99%), HPV33 (24.37%), HPV31 (23.35%), HPV59 (21.87%), HPV39 (19.54%), HPV68 (16.61%) and HPV45 (15.05%). The prevalence of multiple and single HPV persistent infection were 48.66% and 36.71%, respectively; the prevalence of persistent HPV infection in different age groups (<30, 30∼39, 40∼49, >50) were 29.83%, 28.39%, 22.24% and 30.22%, respectively. The follow-up time was significantly associated with heterogeneity by subgroup analysis (P < 0.05), and the prevalence of persistent infection decreased with longer follow-up time.

Conclusions

Multiple infections were more likely to occur persistent HPV infection than single infection. In addition to HPV vaccination, we should emphasize the follow-up management for women under 30 and over 50 years old, those with high-risk HPV infection (HPV59, 39, 68) and multiple infections.

Introduction

Cervical cancer (CC) is a common malignant tumor, and ranks the fourth in the incidence of female malignant tumors in the world (Sung et al., 2021). The etiology of CC is clear, preventable and controllable, effective screening can control the incidence of CC (Van Dyne et al., 2018). Persistent high risk-human papillomavirus (HR-HPV) infection is the main cause of the development of CC (Sinno et al., 2014).

HR-HPV infection is common, especially in sexually active young women, but most infections are transient, spontaneous, and have no clinical symptom. However, 10% of women have persistent HR-HPV infection and are at risk of CC and its precursors (Huh et al., 2015). Persistent HPV infection can increase the risk of high-grade squamous intraepithelial lesions (HSIL) by 250 times (Dalstein et al., 2003). Therefore, adequate attention should be given to persistent HPV infection. To date, the studies mostly focus on the factors that contribute to persistent HPV infection (Rositch et al., 2013; Yang & degree, 2021). However, there is no consensus on which genotypes of persistent HPV infection are most likely to occur. Age is considered a major factor in relation to HPV infection, but the relationship between the age and persistent HPV infection is still controversial.

Studying specific genotypes of persistent HPV infection in clinical practice is crucial for reducing HSIL and guiding CC prevention and treatment. Therefore, we analyzed the prevalence of persistent HPV infection in female worldwide to optimize prevention strategies for CC and provide references for the development of HPV vaccines.

Materials and Methods

The present meta-analysis was performed following the guidelines in the Preferred Reporting Items for Systematic Reviews and Meta-Analysis (PRISMA) statement (Moher et al., 2009). The PRISMA 2009 flow diagram is attached in Fig. 1. Our protocol was registered with the International Prospective Register of Systematic Reviews (PROSPERO) under the code of CRD42022339057.

Figure 1 Preferred reporting items for systematic reviews and meta-analyses flow diagram to search and identify included studies.

Data source

From inception to July 1, 2023, two researchers (MZ and DZ) independently conducted a literature search in Chinese National Knowledge Infrastructure database (CNKI), the Wan Fang database, the Chongqing VIP database, PubMed, Embase and Cochrane library by using a combination of search terms related to human papillomavirus and persistent infection. The full search strategies for different databases were presented in Supplementary Data S1.

Selection criteria

Inclusion criteria

(1) Any HPV infection that continued for at least six months with the same genotype was classified persistent HPV infection. (2) The studies must provide the number of persistent infection and the number of positives at least three genotypes of HR-HPV. (3) All participants in the study were those who tested positive for HPV. (4) The language was limited to Chinese and English.

Exclusion criteria

(1) Review, conference reports, etc. (2) Data was incomplete, or could not be calculated or obtained by contacting the authors. (3) The participants who were pregnant, or those who underwent cervical operation, or those with cervical lesions or other microbial infections. (4) Repeated studies. (5) Studies with quality score <5 based on the AHRQ scale. (6) In the case of some studies based on the same population, only the study reporting the most detailed study was included.

Data extraction

Two investigators (MZ and DZ) independently screened the studies and extracted information. In case of disagreements, they were resolved through discussion or by a third investigator (LML). Each study mainly excerpts the following information: (1) Basic information of included studies: authors, publication time and the research area. (2) Baseline characteristics of each study: the number of HPV positive cases, the positive numbers of HPV genotypes, the positive numbers of single/multiple infection and positive number by age group. (3) Outcome indicators: persistent HPV infection prevalence, the genotypes of persistent HPV infection prevalence, the persistent HPV infection prevalence in each age group, single/multiple persistent HPV infection prevalence.

Quality assessment

Risk of bias was assessed separately by two reviewers (MZ and DZ) as recommended by PRISMA. The quality of studies was assessed by the Agency for Healthcare Research and Quality (AHRQ) checklist. The AHRQ cross-sectional evaluation scale was used to evaluate 11 items with a total score of 11. The higher the total score, the higher the quality of the study. The quality of the study was assessed as follows: low quality = 0∼3; moderate quality = 4∼7; and high quality = 8∼11. Detailed information on quality assessment and risk bias assessment was provided in Table S1.

Statistical analysis

R 4.1.2 (R Foundation for Statistical Computing) was used for the pooled single prevalence of meta-analysis. The pooled prevalence of persistent HPV infection estimates was based on the random-effects model, which gave an overall estimate across studies weighted by sample size, taking the heterogeneity between studies into account (DerSimonian & Laird, 2015). We calculated the pooled effect sizes, along with their respective 95% confidence intervals (CI). Moreover, the heterogeneity between studies was evaluated with the I2 index. If I2 ≧ 50% or P < 0.05, the heterogeneity was considered to be significant. When heterogeneity was significant, subgroup analysis was conducted to explore the potential moderating factors for heterogeneity. Finally, we performed subgroup analysis of the study region, year of publication, sample number, quality scores, follow-up time and the source of sample. Finally, publication bias was analyzed by funnel plot, Egger’s, and Begg’s test. SPSS 26 (Armonk, NY: IBM Corp) was used to perform χ2 test to find the differences of the persistent HPV infection prevalence in different age groups, and we considered P < 0.05 to be significant. To calculate the prevalence of persistent HPV infection, divide the number of persistent HPV infections by the total number of positive cases (Ingabire et al., 2018; Liu et al., 2020).

Results

Characteristic of included literature

Using the search strategies, 10,129 studies were identified, and 2,377 duplicates were excluded. After screening the titles and abstracts, 7,700 unqualified studies were eliminated by assessing and reading the full text of each article. Of the remaining 52 studies, further screening was conducted based on the inclusion and exclusion criteria. Finally, 28 studies were selected and illustrated in Fig. 1. The studies were conducted in 12 different countries, most of the studies were done in China (15), Denmark (2), Netherlands (2), India (1), Brazil (1), Italy (1), Britain (1), Colombia (1), Canada (1), Ghana (1), United States (1) and South Korea (1).

The quality of each study included in the study was evaluated according to the AHRQ. Quality scores of the studies ranged from 5 to 9, with an average of 7, which was shown in Table 1. All of the studies were considered adequate for inclusion in this meta-analysis. Ten studies had a score of ≥ 8, indicating high-quality studies, and 18 studies had a score of 4∼7, indicating medium-quality studies. The characteristics of the selected studies was summarized in Table 1.

Table 1 Basic characteristics of the included literature.

Author	Country	Continent	HPV positive number	Case	Follow-up time (months)	Quality score	
Xu et al. (2021)	China	Asia	488	132	24	6	
Jin & Li (2020)	China	Asia	420	–	12	8	
Wei (2019)	China	Asia	1,586	292	12	6	
Zhong et al. (2018)	China	Asia	340	84	12	7	
Xu et al. (2018)	China	Asia	1,633	347	33	9	
Shen et al. (2018)	China	Asia	704	378	12	8	
Wang et al. (2014)	China	Asia	285	74	12	7	
Hu, Ren & Zhang (2021)	China	Asia	585	163	12	6	
Zhang et al. (2015)	China	Asia	760	172	36	6	
Long et al. (2020)	China	Asia	2,784	564	24	7	
Li et al. (2021)	China	Asia	10,133	4,334	24	7	
Ingabire et al. (2018)	South Korea	Asia	105	13	24	8	
Liu et al. (2020)	China	Asia	565	125	12	8	
Nielsen et al. (2010)	Denmark	Europe	1,166	314	24	7	
Sammarco et al. (2013)	Italy	Europe	55	27	20	6	
Stensen et al. (2016)	Denmark	Europe	2,874	901	54	8	
Li et al. (2017)	China	Asia	85	29	12	8	
Miranda et al. (2013)	Brazil	South America	89	53	24	6	
Cuschieri et al. (2005)	Britain	Europe	126	29	36	5	
Sycuro et al. (2008)	America	North America	147	24	36	6	
Schmeink et al. (2011)	Netherland	Europe	235	–	25.3	8	
Muwonge et al. (2020)	India	Asia	291	–	10	5	
Ye et al. (2010)	China	Asia	400	218	14	8	
Soto-De León et al. (2014)	Colombia	South America	219	–	24	8	
Lai et al. (2008)	China	Asia	412	140	23	7	
Richardson et al. (2003)	Canada	North America	124	69	12	7	
Bulkmans et al. (2007)	Netherland	Europe	620	217	18	7	
Krings et al. (2019a)	Ghana	Africa	104	7	48	7	

Overall prevalence of persistent HPV infection

The total number of positive results was 26,170, and 8,706 of them were persistent HPV infection. The overall pooled persistent HPV infection prevalence was 29.37% (95% CI = [24.05∼35.31]), and the forest plot was shown in Fig. 2.

Figure 2 Forest map of persistent HPV infection prevalence.

Table 2 Results of meta-analysis on persistent HPV infection prevalence.

HPV subtypes	Persistent infection prevalence % (95% CI)	
	Global	Asia	Europe	
HPV16	35.01[29.86∼41.06]	31.65[26.36∼38.01]	40.21[30.15∼50.26]	
HPV52	28.19[23.15∼34.34]	29.99[23.50∼38.28]	24.06[16.55∼32.37]	
HPV58	27.06[20.31∼33.81]	32.92[25.91∼39.92]	22.52[18.46∼27.46]	
HPV18	25.99[19.92∼32.50]	27.59[21.22∼33.97]	29.53[19.81∼41.53]	
HPV33	24.37[17.86∼31.53]	22.99[17.14∼30.10]	33.30[18.87∼47.73]	
HPV31	23.35[16.58∼30.89]	19.28[13.18∼26.23]	38.03[24.82∼53.29]	
HPV59	21.87[13.42∼31.71]	24.46[14.00∼36.74]	20.25[10.37∼39.56]	
HPV39	19.54[13.89∼25.90]	18.38[12.30∼26.55]	22.55[7.70∼42.35]	
HPV68	16.61[11.52∼22.29]	19.21[13.62∼25.41]	13.05[5.43∼23.33]	
HPV45	15.05[9.80∼20.93]	14.81[8.57∼22.40]	26.27[16.11∼42.85]	

Prevalence by genotype

The global prevalence of persistent HPV infections by genotypes was HPV16 (35.01%), HPV52 (28.19%), HPV58 (27.06%), HPV18 (25.99%), HPV33 (24.37%), HPV31 (23.35%), HPV59 (21.87%), HPV39 (19.54%), HPV68 (16.61%), and HPV45 (15.05%).

The prevalence of persistent HPV infection in different continents has been established. The top five genotypes in Asia were HPV58 (32.92%), HPV16 (31.65%), HPV52 (29.99%), HPV18 (27.59%), and HPV59 (24.46%). The top five genotypes in Europe were HPV16 (40.21%), HPV31 (38.03%), and HPV33 (33.30%), HPV18 (29.53%), and HPV45 (26.27%), as shown in Table 2.

Prevalence by multiple/single infection

The prevalence of multiple persistent HPV infections was 48.66% (95% CI [9.80∼87.52]), and the single HPV infection prevalence was 36.71% (95% CI [18.54∼57.05]). Compared with single HPV infection, the prevalence of multiple persistent HPV infections was higher (P < 0.05), and the prevalence by multiple/single infections was listed in Table 3.

Table 3 Results of single/multiple persistent HPV infection prevalence.

HPV infection status	Study (n)	Positive number	I 2 (%)	Results of meta-analysis	
				Persistent infection prevalence (%)	95% CI (%)	
Multiple infections	3	163	92.7	48.66	9.80∼87.52	
Single infection	3	771	94.0	36.71	18.54∼57.05	

Prevalence by age

Five studies evaluated the age-specific prevalence, the infection prevalence in these age groups (<30 years, 30∼39 years, 40∼49 years, >50 years) were 29.83%, 28.39%, 22.24% and 30.22%, respectively. The results of persistent HPV infection prevalence at different age groups were shown in Table 4.

Table 4 Results of persistent HPV infection prevalence at different ages.

Age	Study (n)	Positive number	Persistent infection prevalence (%)	χ 2	P	
<30	4	2870	29.83	15.30	0.002	
30∼40	4	1208	28.39	
40∼50	5	643	22.24	
>50	5	321	30.22	

Subgroup analysis

The heterogeneity test showed that there was significant heterogeneity among the studies (I2 = 98.2%, P < 0.01), so the random effect model was used in the meta-analysis. Subgroup analysis was conducted according to the area, year of publication, sample number, quality score, follow-up time and the source of sample to explore the heterogeneity. As a rule, at least three studies should be available per subgroup. The results showed that the follow-up time might be the sources of heterogeneity (P < 0.05). The prevalence of persistent HPV infection decreased gradually with longer follow-up time, as shown in Fig. 3.

Figure 3 Results of the subgroup analyses to estimation of the prevalence of persistent HPV infection worldwide.

Sensitivity analysis

According to the leave-one-out sensitivity analysis, the pooled prevalence of persistent HPV infection was relatively stable, as shown in Fig. 4.

Figure 4 Results of the forest plot to estimation of the prevalence of persistent HPV infection worldwide based on a random-effects model.

Publication bias

We assessed the publication bias by funnel plot, Egger’s, and Begg’s test. The funnel plot was symmetric, as shown in Fig. 5. No publication bias was found according to the results of both Egger’s (P = 0.085) and Begg’s test (P = 0.399). The p-value in the Egger’s test seemed borderline probably because of the small number of studies. The results of Begg’s and Egger’s tests was consistent, indicating that there was no publication bias in this study.

Figure 5 Result of the funnel plot to estimate persistent HPV infection prevalence worldwide.

Discussion

This systematic review and meta-analysis presented the most recent information about the distribution and prevalence of persistent HPV infection in women globally (Ingabire et al., 2018; Liu et al., 2020), which would provide evidence for the screening, diagnosis, treatment of CC and the development of HPV vaccines.

Persistent infection with HR-HPV is the primary cause of cervical precancerous lesions or CC (Rodriguez et al., 2008; Trottier & Franco, 2006). More than 200 genotypes of HPV have been recognized, of which more than 40 can infect the genital tract (Barros et al., 2020). HPV infection is common, especially in young women, and the majority (∼90%) of newly acquired HPV infection frequently showed a transient course, a phenomenon routinely described as “viral clearance” (Schiffman et al., 2007). Studies have shown that persistent HPV infection varied significantly across different regions (Liu et al., 2019; Ramanakumar et al., 2016). The prevalence of persistent HR-HPV infection was 36.1% in the United States (Bennett et al., 2015) and 26.9%∼38.8% in Europe (Nielsen et al., 2010; Plummer et al., 2007; Schmeink et al., 2013). The study showed that the persistent HPV infection prevalence was 28.38%, which was linked with 27.86% in Shandong, China (Hu, Ren & Zhang, 2021) and 31.40% in Denmark (Stensen et al., 2016), which was lower than a global meta-analysis in 2013 (43%) (Rositch et al., 2013), and it was higher than Korea (12.40%) (Ingabire et al., 2018), the difference might due to the target population’s risks of persistent HPV infection, along with the gap between baseline and the intervals of follow-up. At present, the international definition of persistence HPV infection is not unified, and the duration of persistent infection was controversial. Subgroup analysis showed that the follow-up time was the source of heterogeneity. The previous studies have shown that lower prevalence of persistent HPV infection detected in studies with intervals of 12 months or more compared to studies with intervals of 6 months or less (Rositch et al., 2013).

The distribution of HPV varied greatly in different geographic regions among different ethnic groups (Xu et al., 2021). The study revealed that the most common genotype of persistent HPV infection was HPV 16, which was followed by HPV 58, 52, 18, 33, 31, 59, 39, 68, 45, these results were different from the previous studies (Liang, Chen & Zhang, 2017). Liu et al. (2020) found that HPV58 and53 were the most persistent genotypes, followed by HPV52, 16 and 39. The ranking of persistent HPV infection was HPV16, 18, 33, 31, 52, 39, 56, 45, 58, 35, 68, 51, 66 in 2013 (Rositch et al., 2013). In Asian populations, HPV52 and HPV58 were more common, especially in China (Bee et al., 2021; Yi et al., 2022). In China, HPV52 (21.7%) was the most common genotype, followed by HPV58 (18.2%) (Zhao et al., 2019), while the prevalence of HPV52 and HPV58 was lower in Sweden (Lagheden et al., 2018). The prevalence of HPV45 in European countries was relatively high, especially 7% in Sweden (Lagheden et al., 2018). A global study showed the prevalence of HPV45 was 11.6% (Pirog et al., 2014). However, the prevalence of HPV45 was low in Asia, only 0.5% in Guangdong, China (Jing et al., 2014) and 2.2% in India (Munjal et al., 2014). Most of the study population were from Asia, which might contribute to the lower persistent infection prevalence of HPV45, but higher persistent infection prevalence of HPV52 and 58 in current study. Epidemiological evidence has confirmed that HPV carcinogens are mainly HPV16, 18, 31, 33, 35, 39, 45, 51, 52, 56, 58 and 59. HPV68 is a possible carcinogen because it can transform infected cells into malignant tumor cells (Manini & Montomoli, 2018). These genotypes are called high-risk (World Health Organization, 2017; De Martel et al., 2017). Although HPV vaccines have been proved to prevent most genotypes of HPV infection (De Oliveira, Fregnani & Villa, 2019), currently available 9-valent HPV vaccine do not fully cover all genotypes of HPV related diseases (De Martel et al., 2017). Molina-Pineda et al. (2020) found the prevalence of HPV59 reached 11.5% in CC patients, and it was also found in the top five genotypes of HPV infection detected in different regions of the world, such as Ghana (Krings et al., 2019b), China (Ye et al., 2015) and Switzerland (Dobec et al., 2009). 31% cervical intraepithelial neoplasia I (C1N I) and 26% cervical intraepithelial neoplasia II (CIN II) or above were ascribed to HPV51, HPV53, HPV56, HPV68, and among the 14 HPV genotypes did not covered by the 9-valent vaccine, HPV68 had a higher infection prevalence (9.3%) (Ma et al., 2022). A higher prevalence of HPV68 was also found in non-vaccine in 2021 (Schlecht et al., 2021). Due to the high prevalence of persistent infection and carcinogenesis of HPV 59, 39 and 68 in the biological importance of invasive CC, it was recommended to include them in the next generation of preventive HPV vaccines. Also, to prevent CC, it was necessary to reinforce the follow-up and detect cervical lesions at an early stage, as well as to extend the duration of clinical intervention.

This study found that multiple HPV infections were more likely to occur persistent infection than single infection, which was linked with the previous studies (Van der Weele et al. 2016; Yang & degree, 2021), the reasons were related to the synergistic effects between different genotypes of HPV infection in multiple infections. The limited number of studies included created wide confidence intervals. It is hoped that more samples will be included for evaluation in the future.

To date, the relationship between age and the persistent HPV infection is still controversial. Some studies have reported that lower age was related with increased risk of persistence infection (Rijkaart et al., 2006; Rosa et al., 2008). Additionally, other studies found no association between persistent HPV infection and age (Munoz et al., 2009; Trottier et al., 2008). The patients under the age of 30 and those over 50 should receive special attention during follow-up since they were more likely to have persistent HPV infection. These women under 30 were highly affected by persistent infection, which might be due to their active sexual activity. Indeed, the immature cervix did not produce enough cervical mucus which can increase the risk of HPV infection (Tounkara et al., 2021). One possible explanation for the high persistent infection prevalence in women over 50 years old was that women’s immune function gradually weakens with time, which might lead to HPV escape from the host immune system (Gonzá lez et al., 2010). This was in line with some studies that women over the age of 50 have a high prevalence of persistent infection (Zhang et al., 2015).

There were some limitations in the study. First, there was substantial heterogeneity of the included studies. Despite the fact that heterogeneity is often unavoidable when conducting meta-analyses of observational studies, it does not necessarily mean that the results were invalid (Noubiap et al., 2019). Second, the distribution of genotypes in other regions could not be studied, because of the limited data on the prevalence of persistent infection of specific subtypes in continents, other than Asia and Europe. Third, due to the limited studies on specific single / multiple persistent infections, the specific genotypes of multiple infections that were prone to occur persistent HPV infection have not been studied yet. Additionally, the relationship between the genotypes of persistent infection and the grade of cervical lesions will be the focus of our next study.

It was widely accepted that persistent HR-HPV infection was a major factor in the progression of CC, making it necessary to have extensive screening programs, particularly for women under 30 and over 50 years old, women with high-risk HPV infection (HPV59, 39, 68) and multiple infections. Despite its limitations, the study has important implications for clinical screening of CC and the development of HPV vaccine.

Conclusion

The study would provide a basis for the development of CC screening strategies and HPV vaccines. In addition to HPV vaccination, we should emphasize the follow-up management for women under 30 and over 50 years old, women with high-risk HPV infection (HPV59, 39, 68) and multiple infections.

Supplemental Information

Supplemental Information 1 Data

Click here for additional data file.

Supplemental Information 2 Code

Click here for additional data file.

Supplemental Information 3 Agency for Healthcare Research and Quality for assessing the quality of studies in the meta-analysis

Click here for additional data file.

Supplemental Information 4 Search strategy

Click here for additional data file.

Supplemental Information 5 PRISMA checklist

Click here for additional data file.

Supplemental Information 6 PROSPERO protocol registration

Click here for additional data file.

Supplemental Information 7 Rationale for systematic and meta-analysis conducted

Click here for additional data file.

Thanks Dr. Qian Sun and Pro Rongqiang Zhang for the initial comments on the paper and for editing the final draft.

Additional Information and Declarations

Competing Interests

Author Contributions

Data Availability

The authors declare there are no competing interests.

Ming Zhao conceived and designed the experiments, authored or reviewed drafts of the article, and approved the final draft.

Dan Zhou conceived and designed the experiments, authored or reviewed drafts of the article, and approved the final draft.

Min Zhang performed the experiments, analyzed the data, prepared figures and/or tables, and approved the final draft.

Peipei Kang performed the experiments, analyzed the data, prepared figures and/or tables, and approved the final draft.

Meimei Cui analyzed the data, prepared figures and/or tables, and approved the final draft.

Liling Zhu conceived and designed the experiments, authored or reviewed drafts of the article, and approved the final draft.

Limei Luo conceived and designed the experiments, authored or reviewed drafts of the article, and approved the final draft.

The following information was supplied regarding data availability:

The raw measurements are available in the Supplementary Files.

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
