# Peer review of "Characteristic of persistent human papillomavirus infection in women worldwide: a meta–analysis"

_PeerJ, doi:10.7717/peerj.16247_

## Round 0.1 · original submission · Minor Revisions

R1 has asked for substantial revision of the manuscript; but has highlighted the importance of such a study. I would be willing to go through the revised manuscript and make a quick decision.

·

Basic reporting

The introduction provides a solid foundation regarding cervical cancer (CC) and its relationship with high-risk-human papillomavirus (HR-HPV) infections. The materials and methods section is comprehensive, detailing the PRISMA guidelines, databases used, and the selection criteria for included studies. The exhaustive results section presents detailed figures and statistics based on the literature analyzed. The discussion interprets these findings in the context of previous research and the global state of HPV infections.

Experimental design

The meta-analysis was rigorously conducted, adhering to the PRISMA guidelines. A robust and diverse range of databases was consulted, ensuring a comprehensive literature search. The inclusion and exclusion criteria are specific, clear, and relevant to the study's focus. Registering the protocol with PROSPERO also adds credibility to the methodology.

Validity of the findings

The authors analyzed a significant amount of literature (28 out of 10,129) from various countries. This broad sampling provides a reasonably representative overview of global persistent HPV infection prevalence. The quality scores of the selected studies indicate an average-to-high quality of the sources, giving further weight to the findings. The heterogeneity test results suggest that the authors correctly considered the variations between studies. While some sources of heterogeneity were identified, the robust sensitivity analysis and lack of publication bias strengthened the validity of the findings.

Additional comments

Introduction
The introduction overall provides context and importance on the topic of cervical cancer and HR-HPV infections. However, several points can be refined for better clarity and coherence. Here are the suggested improvements:
1. The claim that cervical cancer "is the only cancer with definite etiology" (lines 39-40) is vital. This should be clarified or expanded upon, as many cancers have known etiologies.
2. The phrase "HR-HPV is inclined to bring about a persistent infection that has reached an agreement" (lines 45-46) is ambiguous. It is unclear what "reached an agreement" means in this context.
3. Establish a more coherent flow when transitioning between different facts and ideas. For instance, before jumping into HR-HPV's inclination for persistent infection, you could first briefly touch upon the significance of understanding persistent infections.
4. "However, there is no agreement on the specific subtypes of persistent HPV infection was prone to occur."(lines 47-48) This sentence is somewhat confusing. It can be more explicit if phrased: "However, there is no consensus on which specific subtypes of persistent HPV infection are most likely to occur."
5. When mentioning that the relationship between age and persistent HPV infection is controversial, it might help to briefly explain why this is relevant or significant to the larger conversation.
6. The introduction can be more concise in certain parts. For instance, "In clinical practice, the study of the specific genotype of HR-HPV persistent infection is of great significance to reduce the incidence of high-grade lesions, which can be a helpful guide for the prevention and treatment of CC." (Line 50-52) can be trimmed to: "Studying specific genotypes of HR-HPV persistent infection in clinical practice is crucial for reducing high-grade lesions and guiding CC prevention and treatment."
7. Make the study's main objective more transparent at the end of the introduction. The current phrasing does establish the purpose but could be made more direct.
8. Ensure consistency in terminology. If "persistent HPV infection" and "HR-HPV persistent infection" mean the same thing, stick to one term for clarity.
Materials and Methods
The "Materials and Methods" section appears detailed, but there are areas where clarity, redundancy, or organization can be improved. Here are some suggestions for enhancement:
General Comments:
1. Structure: Group related items together. For example, all the information on the database search should be together, then selection criteria, screening, extraction, etc. This will make the section more intuitive.
2. Redundancy: Avoid repetition. The repeated statement about the quality of the literature being higher with a higher score can be reduced to just one mention. (Line 115-116)
3. Clarify Terms: Make sure all terms and acronyms are clearly defined the first time they appear.
Specific Comments:
Data Source (Line 64-79)
• Specify that the "search terms" are for English databases, and clarify if a different search strategy was used for the Chinese databases. If so, provide those terms.
• It might be clearer to present the search strategy in a box or supplementary material rather than in the text to keep the flow smoother.
Selection Criteria:
• Under "Inclusion Criteria," clarify point (3) (Line 85-86). Does "those who tested positive for HPV" mean only studies where all participants were HPV positive, or where some were?
• For "Exclusion Criteria," be explicit about the meaning of "quality score <5." Is this based on the AHRQ scale? If so, mention that.
Literature Screening and Data Extraction:
• Clarify "Outcome indicators" (Line 104). Is this what the investigators were extracting from each study? Make this explicit.
Quality Evaluation of the Literature:
• The statement "The higher the total score, the higher the quality of the literature" is repeated twice. Remove one.
• Make a table or visual representation for the score ranges (low, moderate, high) for clarity.
Statistical Analysis:
• In the description of the meta-analysis, specify if any specific methods or models were used (e.g., DerSimonian and Laird random-effects model).
• "R 4.1.2 (College Station, Texas, USA)" seems off. R is software for statistical computing from the R Foundation, while College Station, Texas, is where Stata, another statistical software, is headquartered. Confirm and correct this.
• Describe any methods or transformations used to calculate pooled prevalence, especially using a random effects model. I will suggest you rephrase the sentence (Line 120-122)
• Clarify the criteria for deciding when to perform subgroup analysis. Is it based on significant heterogeneity? If so, specify.
General Stylistic and Grammar Suggestions:
• Change "literature" to "studies" or "articles." "literature" is generally used as an uncountable noun in English.
• In the sentence "The protocol was registered with the International...", consider adding "our" before "protocol" to specify that the protocol belongs to the authors.
These points should help to clarify and streamline the "Materials and Methods" section, making it easier for readers to understand and follow the methodology of the meta-analysis.

Results
The results section you provided is already detailed, but there's room for improvement for clarity and ease of understanding. Here are some suggestions:
1. Characteristic of Included Literature: Provide a breakdown of the number of studies per country to give readers an idea of each country's weight in the meta-analysis.
2. Overall Prevalence of Persistent HPV Infection: Improve readability: Separate different data points with bullet points or subheadings, such as “Total Number of Positive Results,” “Overall Prevalence,” and “Prevalence by Genotype.”
3. Prevalence by Continent & Genotype
 Line 158-162. It is unclear which countries in Asia and Europe have their HPV genotypes prevalence listed. I am wondering if there were no studies in Africa and America.
4. Prevalence by Multiple/Single infection
 Given the wide confidence intervals, especially for multiple persistent HPV infections, briefly discuss possible reasons or the implications of such wide variability.
5. Prevalence by Age
 Discuss or hypothesize why the infection prevalence is higher in the <30 years and >50 years age groups compared to the 30-49 years range.
6. Sources of Heterogeneity, Sensitivity Analysis, and Literature Bias
 Use bullet points or subheadings for clarity, such as “Heterogeneity,” “Subgroup Analysis,” “Sensitivity Analysis,” and “Publication Bias.”
 Given the high I² value (98.2%), the authors should discuss potential reasons for such high heterogeneity and the implications for the meta-analysis findings.
 Mentioning that there's no publication bias according to Egger’s test is excellent, but the P-value provided (P=0.085) seems borderline. It would be beneficial to provide a brief discussion about this.
7. General Suggestions
 Use consistent language: For instance, use “literature” or “studies” consistently throughout the section.
 Check grammar and phrasing: For example, “Characteristics of Included Literature” might be better than “Characteristics of the Included Studies.”
 Avoid repetition: Phrases like “The higher the total score, the higher the quality of the literature” are repeated and can be condensed for brevity.
While the section contains a lot of valuable data, breaking down the information into more digestible chunks and providing more explicit context or explanations will make it more accessible for readers.

Discussion
Here are some comments and suggestions to improve the discussion section:
1. Start with a succinct introductory sentence to reiterate the aim and significance of the study.
2. Avoid repeating the same information. The authors mention that HPV16 is the most common genotype multiple times. It would be more effective to mention it once with comprehensive detail.
3. When comparing the study's results to previous literature, ensure that differences are highlighted and potential reasons for these differences are proposed.
4. Be cautious with making causal claims or speculations, especially without robust evidence. Phrases like "frequent or inappropriate sexual behavior" might be considered judgmental and should be replaced with more neutral language.
5. Emphasize the clinical or public health implications of your findings. E.g., the importance of targeting specific genotypes in vaccines or focusing on particular age groups for screening.
6. Group similar themes together for better flow. For instance, discussion about the differences in HPV genotype prevalence across countries could be clustered together before moving to the implications for vaccines.
7. The assertion that younger women's reproductive systems are still developing and lack time to clear HPV needs references and further clarification. (Lines 254-257)
8. It's great that limitations are acknowledged. Consider discussing how these limitations might impact the study's findings and any implications for future research. (Lines 262-270)

·

Basic reporting

No comments

Experimental design

No comments

Validity of the findings

No comments

Additional comments

The manuscript titled “Characteristic of persistent Human Papillomavirus infection in women worldwide: A meta–analysis”, authored by Ming Zhao, Dan Zhou, Min Zhang, Peipei Kang, Meimei Cui, Liling Zhu and Limei Luo, submitted to PeerJ summarizes results from a metanalysis study conducted to evaluate the pooled prevalence of persistent infection with single or multiple HPV types among women. The information on specific genotype(s) of persistent HR-HPV infection leading to high-grade lesions is highly significant in guiding prevention/treatment strategies. This becomes even more relevant especially in the context of availability of vaccines to several types. In my opinion, the authors have addressed the right question, applied appropriate methods, and have effectively presented the results in the manuscript. My recommendation would be to “Accept” the manuscript for publication in PeerJ.

---

## Round 0.2 · accepted · Accept

I believe that all the points from R1 have been addressed. The manuscript is now much improved. I am pleased to accept the submission for publication.